# CoDraw: Collaborative Drawing as a Testbed for Grounded Goal-driven Communication

## Abstract

In this work, we propose a goal-driven collaborative task that contains language, vision, and action in a virtual environment as its core components. Specifically, we develop a *Co*llaborative image-*Draw*ing game between two agents, called *CoDraw*. Our game is grounded in a virtual world that contains movable clip art objects. The game involves two players: a *Teller* and a *Drawer*. The Teller sees an abstract scene containing multiple clip art pieces in a semantically meaningful configuration, while the Drawer tries to reconstruct the scene on an empty canvas using available clip art pieces. The two players communicate via two-way communication using natural language. We collect the CoDraw dataset of ∼10K dialogs consisting of ∼138K messages exchanged between human agents. We define protocols and metrics to evaluate the effectiveness of learned agents on this testbed, highlighting the need for a novel *crosstalk* condition which pairs agents trained independently on disjoint subsets of the training data for evaluation. We present models for our task, including simple but effective nearest-neighbor techniques and neural network approaches trained using a combination of imitation learning and goal-driven training. All models are benchmarked using both fully automated evaluation and by playing the game with live human agents.

## 1 Introduction

Building agents that can interact with humans in natural language while perceiving and taking actions in their environments is one of the fundamental goals in artificial intelligence. One of the required components, language understanding, has traditionally been studied in isolation and with tasks aimed at imitating human behavior (*e.g.* language modeling Bengio et al. (2003); Mikolov et al. (2010), machine translation Bahdanau et al. (2014); Sutskever et al. (2014), *etc.*) by learning from large text-only corpora. To incorporate both vision and action, it is important to have the language *grounded* (Harnad, 1990; Barsalou, 1999), where words like *cat* are connected to visual percepts and words like *move* relate to actions taken in an environment. Additionally, judging language understanding purely based on the ability to mimic human utterances has limitations: there are many ways to express roughly the same meaning, and conveying the correct information is often more important than the particular choice of words. An alternative approach, which has recently gained increased prominence, is to train and evaluate language generation capabilities in an *interactive* setting, where the focus is on *successfully communicating information* that an agent must share in order to achieve its goals.

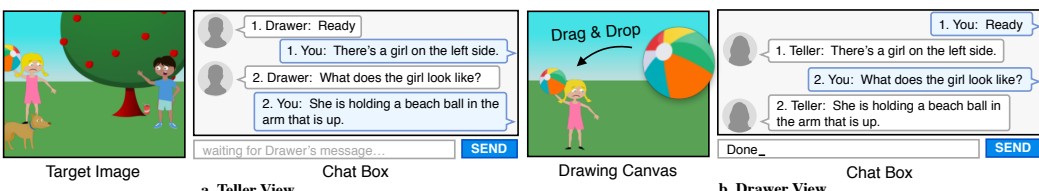

Figure 1: Overview of the proposed Collaborative Drawing (CoDraw) task. The game consists of two players – Teller and Drawer. The Teller sees an abstract scene, while the Drawer sees an empty canvas. Both players need to collaborate and communicate so that the Drawer can reconstruct the image of the Teller by dragging and dropping clip art objects.

In this paper, we propose the Collaborative Drawing (CoDraw) task, which combines grounded language understanding and learning effective goal-driven communication into a single, unified testbed. This task involves perception, communication, and actions in a partially observable virtual environment. As shown in Figure 1, our game is grounded in a virtual world constructed by clip art objects (Zitnick et al., 2013; Zitnick & Parikh, 2013). Two players, Teller and Drawer, play the game. The Teller sees an abstract scene made from clip art objects in a semantically meaningful configuration, while the Drawer sees a drawing canvas that is initially empty. The goal of the game is to have both players communicate so that the Drawer can reconstruct the image of the Teller, without ever seeing it.

Our task requires effective communication because the two players cannot see each other's scenes. The Teller has to describe the scene in sufficient detail for the Drawer to reconstruct it, which will require rich grounded language. Moreover, the Drawer will need to carry out a series of actions from a rich action space to position, orient, and resize all of the clip art pieces required for the reconstruction. Note that such actions are only made possible through clip art pieces: they can represent semantically meaningful configurations of a visual scene that are easy to manipulate, in contrast to low-level pixel-based image representations. The performance of a pair of agents is judged based on the quality of reconstructed scenes. We consider high-quality reconstructions as a signal that communication has been successful.

As we develop models and protocols for CoDraw, we found it critical to train the Teller and the Drawer separately on disjoint subsets of the training data. Otherwise, the two machine agents may conspire to successfully achieve the goal while communicating using a shared "codebook" that bears little resemblance to natural language. We call this separate-training, joint-evaluation protocol *crosstalk*, which prevents learning of mutually agreed upon codebooks, while still checking for goal completion at test time. We highlight crosstalk as one of our contributions, and believe it can be generally applicable to other related tasks (Sukhbaatar et al., 2016; Foerster et al., 2016; de Vries et al., 2016; Das et al., 2017b; Lewis et al., 2017).

**Summary of Contributions**

- We propose a novel CoDraw task, which is a game designed to facilitate the learning and evaluation of effective natural language communication in a grounded context.
- We collect a CoDraw dataset of ∼10K variable-length dialogs consisting of ∼138K messages with the drawing history at each step of the dialog.
- We define a scene similarity metric, which allows us to automatically evaluate the effectiveness of the communication at the end and at intermediate states.
- We propose a cross-talk training and evaluation protocol that prevents agents from potentially learning joint uninterpretable codebooks, rendering them ineffective at communicating with humans.
- We evaluate several Drawer and Teller models automatically as well as by pairing them with humans, and show that long-term planning and context reasoning in the conversation are key challenges of the CoDraw task.

## 2 RELATED WORK

**Language and Vision.** The proposed CoDraw game is related to several well-known language and vision tasks that study grounded language understanding (Karpathy & Fei-Fei, 2015; Donahue et al., 2015; de Vries et al., 2016). For instance, compared to image captioning (Vinyals et al., 2017; Xu et al., 2015; Chen & Zitnick, 2015; Lu et al., 2017), visual question answering (Antol et al., 2015; Zhang et al., 2016; Goyal et al., 2016; Gao et al., 2015; Krishna et al., 2017; Malinowski & Fritz, 2014; Ren et al., 2015; Tapaswi et al., 2016; Yu et al., 2015; Zhu et al., 2016) and recent embodied extensions (Das et al., 2018), CoDraw involves multiple rounds of interactions between two agents. Both agents hold their own partially observable states and need to build a mental model for each other to collaborate. Compared to work on navigation (Vogel & Jurafsky, 2010; Anderson et al., 2018; Fried et al., 2018b) where an agent must follow instructions to move itself in a static environment, CoDraw involves moving and manipulating multiple clip art pieces, which must jointly form a semantically meaningful scene. Compared to visual dialog (Das et al., 2017a;b; Strub et al., 2017; Mostafazadeh et al., 2017) tasks, agents need to additionally cooperate to change the environment with actions (*e.g.*, move around pieces). Thus, the agents have to possess the ability to

adapt and hold a dialog about novel scenes that will be constructed as a consequence of their dialog. In addition, we also want to highlight that CoDraw has a well-defined communication goal, which facilitates objective measurement of success and enables end-to-end goal-driven learning.

**End-to-end Goal-Driven Dialog.** Traditional goal-driven agents are often based on 'slot filling' (Lemon et al., 2006; Wang & Lemon, 2013; Yu et al., 2015), in which the structure of the dialog is pre-specified but the individual slots are replaced by relevant information. Recently, end-to-end neural models are also proposed for goal-driven dialog (Bordes et al., 2017; Li et al., 2017a;b; He et al., 2017), as well as goal-free dialog or 'chit-chat' (Shang et al., 2015; Sordoni et al., 2015; Vinyals & Le, 2015; Li et al., 2016; Dodge et al., 2016). Unlike CoDraw, in these approaches, symbols in the dialog are not grounded into visual objects.

**Language Grounded in Environments.** Learning language games to change the environment has been studied recently (Wang et al., 2016; 2017). The agent can change the environment using the grounded natural language. However, agents do not have the need to cooperate. Other work on grounded instruction following relies on datasets of pre-generated action sequences annotated with human descriptions, rather than using a single end goal (Long et al., 2016). Speaker models for these tasks are only evaluated based on their ability to describe an action sequence that is given to them (Fried et al., 2018a), whereas Teller models for CoDraw also need to select a desired action sequence in a goal-driven manner. Language grounding has also been studied for robot navigation, manipulation, and environment mapping (Tellex et al., 2011; Mei et al., 2015; Daniele et al., 2016). However, these works manually pair each command with robot actions and lack end-to-end training (Tellex et al., 2011), dialog (Mei et al., 2015; Daniele et al., 2016), or both (Walter et al., 2014).

**Emergent Communication.** Building on the seminal works by Lewis (1969; 1975), a number of recent works study cooperative games between agents where communication protocols emerge as a consequence of training the agents to accomplish shared goals (Sukhbaatar et al., 2016; Foerster et al., 2016). These methods have typically been applied to learn to communicate small amounts of information, rather than the complete, semantically meaningful scenes used in the CoDraw task. In addition, the learned communication protocols are usually not natural (Kottur et al., 2017) or interpretable. On the other hand, since our goal is to develop agents that can assist and communicate with humans, we must pre-train our agents on human communication and use techniques that can cope with the greater linguistic variety and richness of meaning present in natural language.

## 3  CODRAW TASK AND DATASET

In this section, we first detail our task, then present the CoDraw dataset, and finally propose a Scene Similarity Metric which allows automatic evaluation of the reconstructed and original scene.

### 3.1  TASK

**Abstract Scenes.** To enable people to easily draw semantically rich scenes on a canvas, we leverage the Abstract Scenes dataset of (Zitnick et al., 2013; Zitnick & Parikh, 2013). This dataset consists of 10,020 semantically consistent scenes created by human annotators. An example scene is shown in the left portion of Figure 1. Most scenes contain 6 objects (min 6, max 17, mean 6.67). These scenes depict children playing in a park, and are made from a library of 58 clip arts, including a boy (Mike) and a girl (Jenny) in one of 7 poses and 5 expressions, and various other objects including trees, toys, hats, animals, food, etc. An abstract scene is created by dragging and dropping multiple clip art objects to any $(x, y)$ position on the canvas. Also, for each clip art, different spatial transformations can be applied, including sizes (Small, Normal, Large), and two orientations (facing left or right). The clip art serve simultaneously as a high-level visual representation and as a mechanism by which rich drawing actions can be carried out.

**Interface.** We built a drag-and-drop interface based on the Visual Dialog chat interface (Das et al., 2017a) (see Figures 4 and 5 in Appendix A for screen shots of the interface). The interface allows real-time interaction between two people. During the conversation, the Teller describes the scene and answers any questions from the Drawer on the chat interface, while Drawer "draws" or reconstructs the scene based on the Teller's descriptions and instructions. Each side is only allowed to send one message at a time, and must wait for a reply before continuing. The maximum length of a

single message is capped at 140 characters: this prevents excessively verbose descriptions and gives the Drawer more chances to participate in the dialog by encouraging the Teller to pause more frequently. Both participants were asked to submit the task when they are both confident that Drawer has accurately reconstructed the scene of Teller. Our dataset, as well as this infrastructure for live chat with live drawing, will be made publicly available.

**Additional Interaction.** We did not allow Teller to continuously observe Drawer's canvas to make sure that the natural language focused on the high-level semantics of the scene rather than instructions calling for the execution of low-level clip art manipulation actions, but we hypothesize that direct visual feedback may be necessary to get the all the details right. For this, we give one chance for the Teller to look at the Drawer's canvas using a 'peek' button in the interface. Communication is only allowed after the peek window is closed.

## 3.2 DATASET

We collect 9,993[1] dialogs where pairs of people complete the CoDraw task, consisting of one dialog per scene in the Abstract Scenes dataset. The dialogs contain of a total of 138K utterances and include snapshots of the intermediate state of the Drawer's canvas after each round of each conversation. We reserve 10% of the scenes (1,002) to form a test set and an additional 10% (1,002) to form a development set; the corresponding dialogs are used to evaluate human performance for the CoDraw task. The remaining dialogs are used for training (see Section 5 for details about our training and evaluation setup.)

The message length distribution for the Drawer is skewed toward 1 with the passive replies like *"ok"*, *"done"*, etc. There does exist a heavy tail, which shows that Drawers do ask clarifying questions about the scene like *"where is trunk of second tree, low or high"*. On the other hand, the distribution of number of tokens in Tellers' utterances is relatively smooth with long tails. The vocabulary size is 4,555. Since the subject of conversations is about abstract scenes with a limited number of clip arts, the vocabulary is relatively small compared to those on real images. See Appendix B for a more detailed analysis of our dataset, where we study the lengths of the conversations, the number of rounds, and the distributions of scene similarity scores when humans perform the task.

## 3.3 SCENE SIMILARITY METRIC

The goal-driven nature of our task naturally lends itself to evaluation by measuring the similarity of the reconstructed scene to the original. For this purpose we define a scene similarity metric, which allows us to automatically evaluate communication effectiveness both at the end of a dialog and at intermediate states. We use the metric to compare how well different machine-machine, human-machine, and human-human pairs can complete the CoDraw task.

Let $c_i, c_j$ denote the identity, location, configuration of two clipart pieces $i$ and $j$. A clipart image $C = \{c_i\}$ is then simply a set of clipart pieces. Given two images $C$ and $\hat{C}$, we compute scene similarity by first finding the common clipart pieces $C \cap \hat{C}$ and then computing unary $f(c_i)$ and pairwise terms $g(c_i, c_j)$ on these pieces in common:

$$s(C, \hat{C}) = \underbrace{\frac{\sum_{c \in C \cap \hat{C}} f(c)}{|C \cup \hat{C}|}}_{\text{unary}} + \underbrace{\frac{\sum_{c_i, c_j \in C \cap \hat{C}, i<j} g(c_i, c_j)}{|C \cup \hat{C}|(|C \cap \hat{C}| - 1)}}_{\text{pairwise}} \tag{1}$$

Using $f(c) = 1$ and $g(c_i, c_j) = 0$ would result in the standard intersection-over-union measure used for scoring set predictions. The denominator terms normalize the metric to penalize missing or extra clip art, and we set $f$ and $g$ such that our metric is on a 0-5 scale. The exact terms $f$ and $g$ are described in Appendix C.

## 4 MODELS

We model both the Teller and the Drawer, and evaluate the agents using the metrics described in the previous section. Informed by our analysis of the collected dataset (see Appendix B), we make three

---

[1]Excluding 27 empty scenes from the original dataset.

modeling assumptions compared to the full generality of the setup that humans were presented with during data collection. These assumptions hold for all models studied in this paper.

**Assumption 1: Silent Drawer.** We choose to omit the Drawer's ability to ask clarification questions; instead, our Drawer models will always answer "ok" and our Teller models will not condition on the text of the Drawer replies. This is consistent with typical human replies (around 62% of which only use a single token) and the fact that the Drawer talking is not strictly required to resolve the information asymmetry inherent in the task. We note that this assumption does not reduce the number of modalities needed to solve the task: there is still language generation on the Teller side, in addition to language understanding, scene perception, and scene generation on the Drawer side. Drawer models that can detect when a clarification is required, and then generate a natural language clarification question is interesting future work.

**Assumption 2: No Peek Action.** The second difference is that the data collection process for humans gives the Teller a single chance to peek at the Drawer's canvas, a behavior we omit from our models. Rich communication is still required without this behavior, and omitting it also does not decrease the number of modalities needed to complete the task. We leave for future work the creation of models that can peek at the time that maximizes task effectiveness.

**Assumption 3: Full Clip Art Library.** The final difference is that our drawer models can select from the full clip art library. Humans are only given access to a smaller set so that it can easily fit in the user interface, while ensuring that all pieces needed to reconstruct the target scene are available. We choose to adopt the full-library condition as the standard for models because it gives the models greater latitude to make mistakes (making the problem more challenging) and makes it easier to detect obviously incorrect groundings.

## 4.1 Rule-Based Nearest-Neighbor Methods

Simple methods can be quite effective even for what appear to be challenging tasks, so we begin by building models based on nearest-neighbors and rule-based approaches.

**Rule-based Nearest-Neighbor Teller.** For our first Teller model, We consider a rule-based dialog policy where the Teller describes exactly one clip art each time it talks. The rule-based system determines which clip art to describe during each round of conversation, following a fixed order that roughly starts with objects in the sky (sun, clouds, airplanes), then objects in the scene (trees, Mike, Jenny), and ends with small objects (sunglasses, baseball bat). The Teller then produces an utterance by performing a nearest-neighbor lookup in a database containing (*Teller utterance*, *clip art object*) pairs, where the similarity between the selected clip art and each database element is measured by applying the scene similarity metric to individual clip art. The database is constructed by collecting all instances in the training data where the Teller sent a message and the Drawer responded by adding a single clip art piece to the canvas. Instances where the Drawer added multiple clip art pieces or made any changes to the position or other attributes of pieces already on the canvas are not used when constructing the nearest-neighbor database. This baseline approach is based on the assumptions that the Drawer's action was elicited by the Teller utterance immediately prior, and that the Teller's utterance will have a similar meaning when copied verbatim into a new conversation.

**Rule-based Nearest-Neighbor Drawer.** This Drawer model is the complement to the rule-based nearest-neighbor Teller. It likewise follows a hard-coded rule that the response to each Teller utterance should be the addition of a single clip art to the scene, and makes use of the same database of (*Teller utterance*, *clip art object*) tuples collected from the training data. Each Teller utterance the agent receives is compared with the stored tuples using character-level string edit distance. The clip art object from the most similar tuple is selected and added to the canvas by the Drawer.

## 4.2 Neural Drawer

In this section, we describe a neural network approach to the Drawer. Contextual reasoning is an important part of the CoDraw task: each message from the Teller can relate back to what the Drawer has previously heard or drawn, and the clip art pieces it places on the canvas must form a semantically coherent scene. To capture these effects, our model should condition on the past history of the conversation and use an action representation that is conducive to generating coherent scenes.

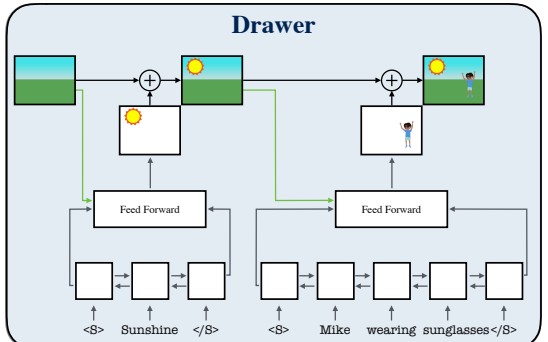 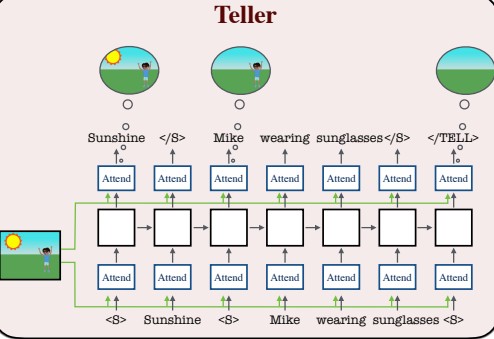

Figure 2: A sketch of our model architectures for the neural Drawer and Teller. The Drawer (left) conditions on the current state of the canvas and a BiLSTM encoding of the previous utterance to decide which clip art pieces to add to a scene. The Teller (right) uses an LSTM language model with attention to the scene (in blue) taking place before and after the LSTM. The "thought bubbles" represent intermediate supervision using an auxiliary task of predicting which clip art have not been described yet. In reinforcement learning, the intermediate scenes produced by the drawer are used to calculate rewards. Note that the language used here was constructed for illustrative purposes, and that the messages in our dataset are more detailed and precise.

When considering past history, we make the Markovian assumption that the current state of the Drawer's canvas captures all information from the previous rounds of dialog. Thus, the Drawer need only consider the most recent utterance from the Teller and the current canvas to decide what to draw next. We experimented with incorporating additional context – such as previous messages from the teller or the action sequence by which the Drawer arrived at its current canvas configuration – but did not observe any gains in performance.

The current state of the canvas is represented using a collection of indicator features and real-valued features. For each of the $n_c = 58$ clip art types, there is an indicator feature for its presence on the canvas, and an indicator feature for each discrete assignment of an attribute to the clip art (e.g. $\mathbf{1}_{\text{size=small}}$, $\mathbf{1}_{\text{size=medium}}$, etc.) for a total of $n_b = 41$ binary features. There are additionally two real-valued features that encode the $x$ and $y$ position of the clip art on the canvas, normalized to the 0-1 range. The resulting canvas representation is a feature vector $v_{\text{canvas}}$ of size $n_c \times (n_b + 2)$, where all features for absent clip art types are set to zero.

We run a bi-directional LSTM over the Teller's most recent message and extract the final hidden states for both directions, which we concatenate to form a vector $v_{\text{message}}$. The Drawer is then a feed-forward neural network that takes as input $v_{\text{canvas}}$ and $v_{\text{message}}$ and produces an output vector $v_{\text{action}}$. The action representation $v_{\text{action}}$ also consists of $n_c \times (n_b + 2)$ elements and can be thought of as a continuous relaxation of the mostly-discrete canvas encoding. For each clip art type, there is a real-valued score that determines whether a clip art piece of that type should be added to the canvas: a positive score indicates that it should be added as part of the action. During training, a binary cross-entropy loss compares these scores with the actions taken by human drawers. $v_{\text{action}}$ also contains unnormalized log-probabilities for each attribute-value assignment (e.g. $z_{\text{size=small}}$, $z_{\text{size=medium}}$, etc. for each clip art type); when a clip art piece is added to the canvas, its attributes are assigned to their most-probable values. The log-probabilities are trained using softmax losses. Finally, $v_{\text{action}}$ contains two entries for each clip art type that determine the clip art's $x, y$ position if added to the canvas; these elements are trained using an $L_2$ loss.

### 4.3 NEURAL TELLER: SCENE2SEQ

For our neural Teller models, we adopt an architecture that we call *scene2seq*. This architecture is a conditional language model over the Teller's side of the conversation with special *next-utterance tokens* to indicate when the Teller ends its current utterance and waits for a reply from the Drawer.[2] The language model is implemented using an LSTM, where information about the ground-truth

---

[2]Though none of the models in this paper handle language in the Drawer replies, these can be incorporated into the *scene2seq* framework similar to the approach of Lewis et al. (2017).

scene is incorporated both before and after each LSTM cell through the use of an attention mechanism. Attention occurs over individual clip art pieces: each clip art object in the ground-truth scene is represented using a vector that is the sum of learned embeddings for different clip art attributes (e.g. $e_{\text{type=Mike}}$, $e_{\text{size=small}}$, etc.) At test time, the Teller's messages are constructed by decoding from the language model using greedy word selection.

To communicate effectively, the Teller must keep track of which parts of the scene it has and has not described, and also generate language that is likely to accomplish the task objective when interpreted by the Drawer. We found that training the *scene2seq* model using a maximum likelihood objective did not result in long-term coherent dialogs for novel scenes. Rather than introducing a new architecture to address these deficiencies, we explore reducing them by using alternative training objectives. To better ensure that the model keeps track of which pieces of information it has already communicated, we take advantage of the availability of drawings at each round of the recorded human dialogs and introduce an auxiliary loss based on predicting these drawings. To select language that is more likely to lead to successful task completion, we further fine-tune our Teller models to directly optimize the end-task goal using reinforcement learning.

### 4.3.1 TRAINING WITH INTERMEDIATE SUPERVISION

We incorporate state tracking into the scene2seq architecture through the use of an auxiliary loss. This formulation maintains the end-to-end training procedure and keeps test-time decoding exactly the same. The only change is that, at each utterance separator token, the output from the LSTM is used to predict which clip art still need to be described. More precisely, the network must classify whether each clip art in the ground truth has been drawn already or not. The supervisory signal makes use of the fact that the CoDraw dataset records human drawer actions at each round of the conversation, not just at the end. The network outputs a score for each clip art ID, which is connected to a softmax loss for the clip art in the ground truth scene (the scores for absent clip arts do not contribute to the auxiliary loss). We find that adding such a supervisory signal reduces the Teller's propensity for repeating itself or omitting objects.

### 4.3.2 FINE-TUNING WITH REINFORCEMENT LEARNING

The auxiliary loss helps the agent be more coherent throughout the dialog, but it is still trained to imitate human behavior rather than to complete the downstream task. By training the agents using reinforcement learning (RL), they can learn to use language that is more effective at achieving high-fidelity scene reconstructions. In this work we only train the Teller with RL, because the Teller has challenges maintaining a long-term strategy throughout a long dialog, whereas preliminary results showed that making local decisions is less detrimental for Drawers.

The scene2seq Teller architecture remains unchanged, and each action from the agent is to output a word or one of two special tokens: a next-utterance token and a stop token. After each next-utterance token, our neural Drawer model is used to take an action in the scene and the resulting change in scene similarity metric is used as a reward. However, this reward scheme alone has an issue: once all objects in the scene are described, any further messages will not result in a change in the scene and have a reward of zero. As a result, there is no incentive to end the conversation. We address this by applying a penalty of 0.3 to the reward whenever the Drawer makes no changes to the scene. We train our model with REINFORCE (Williams, 1992).

## 5 TRAINING PROTOCOL AND EVALUATION

To evaluate our models, we pair our models with other models, as well as with a human.

**Human-Machine Pairs.** We modified the interface used for data collection to allow human-machine pairs to complete the tasks. Each model plays one game with a human per scene in the test set, and we compare the scene reconstruction quality between different models and with human-human pairs.

**Script-based Drawer Evaluation.** In addition to human evaluation, we would like to have automated evaluation protocols that can quickly estimate the quality of different models. Drawer models can be evaluated by pairing them with a Teller that replays recorded human conversation from a script (a recorded dialog from the dataset) and measuring scene similarity at the end of the dia-

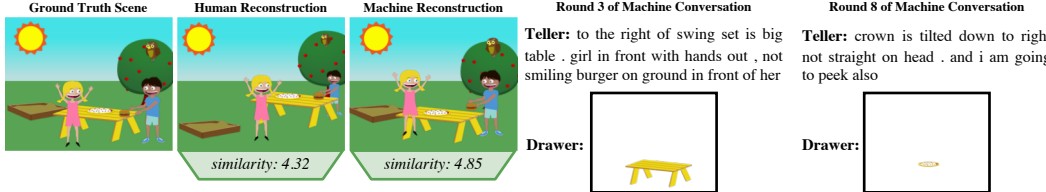

Figure 3: A rule-based nearest-neighbor Teller/Drawer pair "trained" on the same data outperforms humans for this scene according to the similarity metric, but the language used by the models doesn't always correspond in meaning to the actions taken. The three panels on the left show a scene from the test set and corresponding human/model reconstructions. The two panels on the right show the Teller message and Drawer action from two rounds of conversation by the machine agents.

log. While this setup does not capture the full interactive nature of the task, the Drawer model still receives human descriptions of the scene and should be able to reconstruct it. Our modeling assumptions include not giving Drawer models the ability to ask clarifying questions, which further suggests that script-based evaluation can reasonably measure model quality.

**Machine-Machine Evaluation.** Unlike Drawer models, Teller models cannot be evaluated using a "script" from the dataset. We instead consider an evaluation where a Teller model and Drawer model are paired, and their joint performance is evaluated using the scene similarity metric.

### 5.1 CROSSTALK TRAINING PROTOCOL

Automatically evaluating agents, especially in the machine-machine paired setting, requires some care because a pair of agents can achieve a perfect score while communicating in a shared code that bears no resemblance to natural language. There are several ways such co-adaptation can develop. One is by overfitting to the training data to the extent that it's used as a codebook – we see this with the rule-based nearest-neighbor agents described in Section 4.1, where a Drawer-Teller pair "trained" on the same data outperforms humans on the CoDraw task. An examination of the language, however, reveals that only limited generalization has taken place (see Figure 3). Another way that agents can co-adapt is if they are trained jointly, for example using reinforcement learning. To limit these sources of co-adaptation, we propose a training protocol we call "crosstalk." In this setting, the training data is split in half, and the Teller and Drawer are trained separately on disjoint halves of the training data. When multiple agents are required during training (as with reinforcement learning), the joint training process is run separately for both halves of the training data, but evaluation pairs a Teller from the first partition with a Drawer from the second. This ensures to some extent that the models can succeed only if they have learned generalizable language, and not via a highly specialized codebook specific to model instances.

Taking the crosstalk training protocol into account, the dataset split we use for all experiments is: 40% Teller training data (3,994 scenes/dialogs), 40% Drawer training data (3,995), 10% development data (1,002) and 10% testing data (1,002).

### 6 RESULTS

Results for our models are shown in Table 1. All numbers are scene similarities, averaged across scenes in the test set.

**Neural Drawer Performs the Best.** In the script setting, our neural Drawer is able to outperform the rule-based nearest-neighbor baseline (3.39 vs. 0.94) and close most of the gap between baseline (0.94) and human performance (4.17).

**Validity of Script-Based Drawer Evaluation.** To test the validity of script-based Drawer evaluation – where a Drawer is paired with a Teller that recites the human script from the dataset corresponding to the test scenes – we include results from interactively pairing human Drawers with a Teller that recites the scripted messages. While average scene similarity is lower than when using live human Tellers (3.83 vs. 4.17), the scripts are sufficient to achieve over 91% of the effectiveness of the

Table 1: Results for our models on the test set, using three types of evaluation: script-based (i.e. replaying Teller utterances from the dataset), human-machine, and machine-machine pair evaluation.

| | Teller | Drawer | Scene similarity |
|---|---|---|---|
| Script-based | Script (replays human-generated messages) | Rule-based Nearest Neighbor | 0.94 |
| | Script (replays human-generated messages) | Neural Network | 3.39 |
| | Script (replays human-generated messages) | Human | **3.83** |
| Human-Machine | Rule-based Nearest Neighbor | Human | 3.21 |
| | Scene2seq (imitation learning) | Human | 2.69 |
| | + auxiliary loss | Human | 3.04 |
| | + RL fine-tuning | Human | **3.65** |
| Machine-Machine | Rule-based Nearest Neighbor | Neural Network | 3.08 |
| | Scene2seq (imitation learning) | Neural Network | 2.67 |
| | + auxiliary loss | Neural Network | 3.02 |
| | + RL fine-tuning | Neural Network | **3.67** |
| | Human | Human | **4.17** |

same Teller utterances when they were communicated live (according to our metric). The drop in similarity may be in part due to the inability of the Teller to peek at the Drawer's canvas and suggest specific corrections, and in part because the Teller can't answer clarifying questions specific to the Drawer's personal understanding of the instructions. Note that a human Drawer with a script-based Teller still outperforms our best Drawer model paired with a script-based Teller.

**Benefits of Intermediate Supervision and Goal-Driven Training.** Pairing our models with humans shows that the *scene2seq* Teller model trained with imitation learning is worse than the rule-based nearest-neighbor baseline (2.69 vs. 3.21), but that the addition of an auxiliary loss followed by fine-tuning with reinforcement learning allow it to outperform the baseline (3.65 vs. 3.21). However, there is still a gap between to human Tellers (3.65 vs. 4.17). Many participants in our human study noted that they received unclear instructions from the models they were paired with, or expressed frustration that their partners could not answer clarifying questions as a way of resolving such situations. Recall that our Teller models currently ignore any utterances from the Drawer.

**Correlation Between Fully-automated and Human-machine Evaluation.** We also report the result of paired evaluation for different Teller models and our best Drawer, showing that the relative rankings of the different Teller types match those we see when models are paired with humans. This shows that automated evaluation while following the crosstalk training protocol is a suitable automated proxy for human-evaluation.

## 6.1 TYPICAL ERRORS

The errors made by Teller reflect two key challenges posed by the CoDraw task: reasoning about the context of the conversation and drawing, and planning ahead to fully and effectively communicate the information required. A common mistake the rule-based nearest-neighbor Teller makes is to reference objects that are not present in the current scene. Figure 3 shows an example (second panel from the right) where the Teller has copied a message referencing a "swing" that does not exist in the current scene. In a sample of 5 scenes from the test set, the rule-based nearest-neighbor describes a non-existent object 11 times, compared to just 1 time for the scene2seq Teller trained with imitation learning. The scene2seq Teller, on the other hand, frequently describes clip art pieces multiple times or forgets to mention some of them: in the same sample of scenes, it re-describes an object 10 times (vs. 2 for the baseline) and fails to mention 11 objects (vs. 2.) The addition of an auxiliary loss and RL fine-tuning reduces these classes of errors while avoiding frequent descriptions of irrelevant objects (0 references to non-existent objects, 3 instances of re-describing an object, and 4 objects omitted.)

On the Drawer side, the most salient class of mistakes made by the neural network model is semantically inconsistent placement of multiple clip art pieces. Several instances of this can be seen in

Figure 10 in the Appendix D, where the Drawer places a hat in the air instead of on a person's head, or where the drawn clip art pieces overlap in a visually unnatural way.

Qualitative examples of both human and model behavior are provided in Appendix D.

## 7  CONCLUSION

In this paper, we introduce CoDraw: a collaborative task designed to facilitate learning of effective natural language communication in a grounded context. The task combines language, perception, and actions while permitting automated goal-driven evaluation both at the end and as a measure of intermediate progress. We introduce a dataset and models for this task, and propose a *crosstalk* training + evaluation protocol that is more generally applicable to studying emergent communication. The models we present in this paper show levels of task performance that are still far from what humans can achieve. Long-term planning and contextual reasoning as two key challenges for this task that our models only begin to address. We hope that the grounded, goal-driven communication setting that CoDraw is a testbed for can lead to future progress in building agents that can speak more naturally and better maintain coherency over a long dialog, while being grounded in perception and actions.

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

## A  INTERFACE AND DATA COLLECTION

### A.1  INTERFACE

Figure 4 shows the interface for the Teller, and Figure 5 shows the interface for the Drawer. Following previous works (Zitnick et al., 2013; Zitnick & Parikh, 2013), Drawers are given 20 clip art objects selected randomly from the 58 clip art objects in the library, while ensuring that all objects required to reconstruct the scene are available.

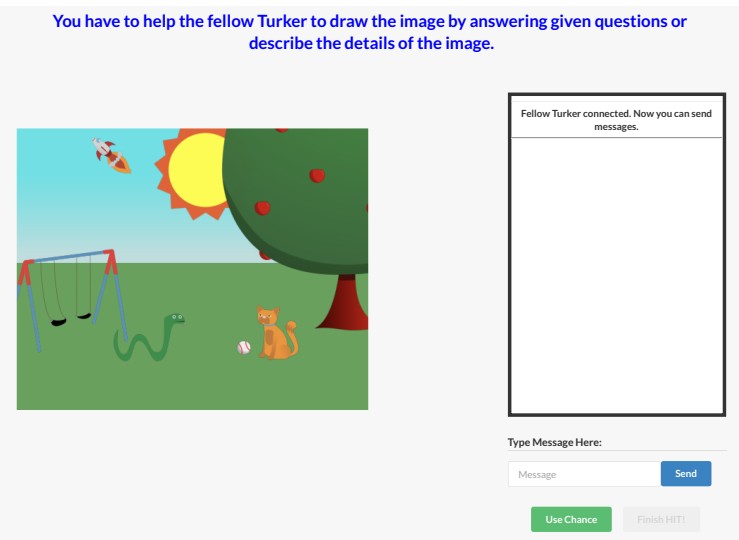

Figure 4: User interface for Teller. The left image is one of abstract scenes from Zitnick & Parikh (2013). The Teller sends messages using an input box. The Teller has a single chance to peek Drawer's canvas to correct mistakes. When the Teller feels good to finish, the Teller can finish the session.

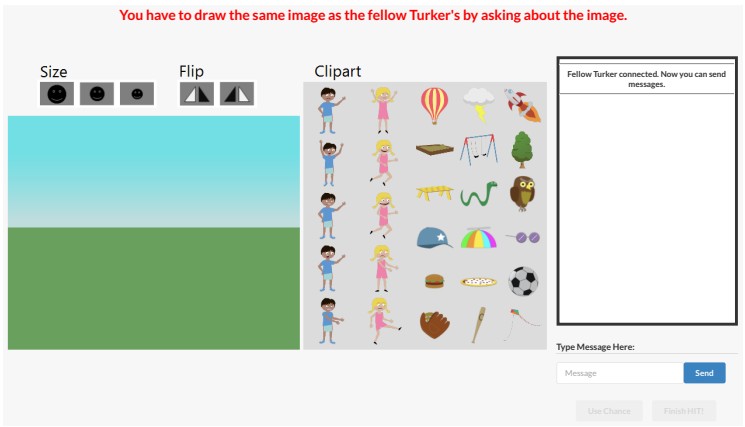

Figure 5: User interface for a Drawer. The Drawer has an empty canvas and a randomly generated drawing palette of Mike, Jenny, and 18 other objects, chosen from a library of 58 clip arts. We ensure that using the available objects, Drawer can fully reproduce the scene. Using the library, the Drawer can draw on the canvas in a drag-and-drop fashion. Drawer can also send messages using a given input box. However, the peek button is disabled. Only the Teller can use it.

## A.2 PARTICIPANT STATISTICS

We found that approximately 13.6% of human participants disconnect voluntarily in an early stage of the session. We paid participants who stayed in the conversation and had posted at least three messages. However, we exclude those incomplete sessions in the dataset, and only use the completed sessions.

There are 616 unique participants represented in our collected data. Among these workers, the 5 most active have done 26.63% of all finished tasks (1,419, 1,358, 1,112, 1,110, and 1,068 tasks). Across all workers, the maximum, median, and minimum numbers of tasks finished by a worker are 1,419, 3, and 1, respectively.

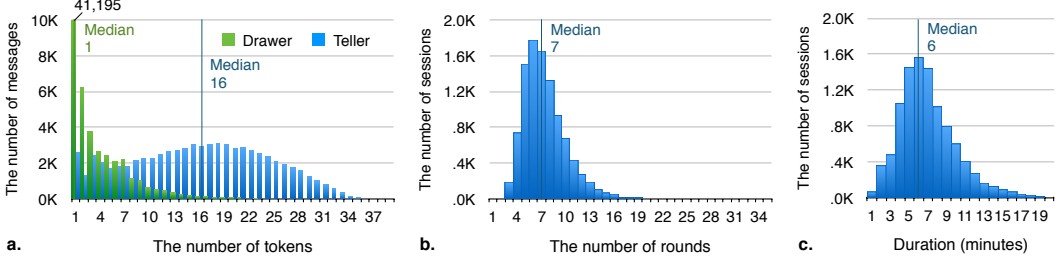

Figure 6: Statistics of the CoDraw dataset. **(a)** The distribution of the number of tokens in Teller's (blue) and Drawer's (green) messages. Notice that the number of single-token messages of Drawer is 41,195 (62.06%). The medians for Teller and Drawer are 16 and 1, respectively. **(b)** The distribution of the numbers of conversational rounds. The median is 7 rounds. **(c)** The distribution of the duration of dialog sessions. The median is 6 minutes.

## B DATASET STATISTICS

The CoDraw dataset consists of 9,993 dialogs consisting of a total of 138K utterances. Each dialog describes a distinct abstract scene.

**Messages.** Figure 6a shows the distribution of message lengths for both Drawers and Tellers. Drawer messages tend to be short (the median length is 1 accounts for 62% of messages), but there does exist a heavy tail where the Drawer asks clarifying questions about the scene. Teller message length have a more smooth distribution with a median length of 16 tokens. The size of vocabulary is 4,555: since conversations describe abstract scenes consisting of a limited number of clip art types, the vocabulary is relatively small compared to tasks involving real images.

**Rounds.** Figure 6b shows the distribution of the numbers of conversational rounds for dialog sessions. Most interactions are shorter than 20 rounds, median being 7.

**Durations.** In Figure 6c we see that the median session duration is 6 minutes. We had placed a 20-minute maximum limit on each session.

**Scores.** Figure 7 shows the distribution of scene similarity scores throughout the dataset. Figure 8 shows the progress of scene similarity scores over the rounds of a conversation. An average conversations is done improving the scene similarity after about 5 rounds, but for longer conversations that continue to 23 rounds, there is still room for improvement.

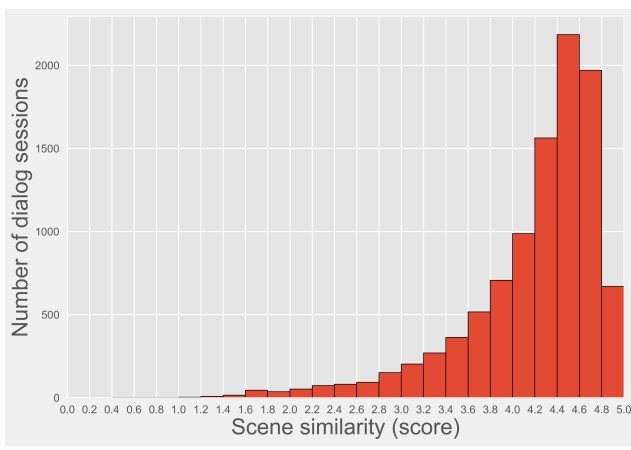

Figure 7: The distribution of overall scores (section 3.3) at the end of the dialog.

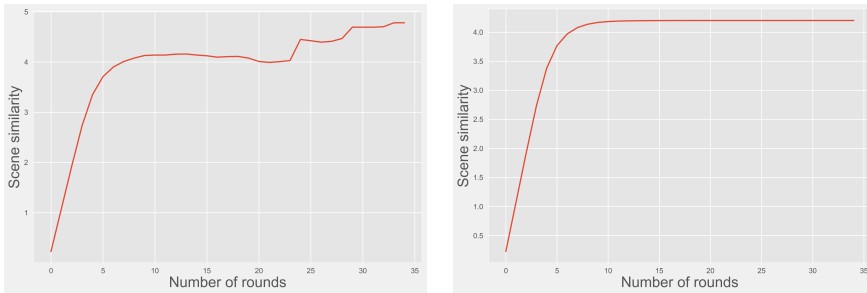

Figure 8: Average scene similarity plotted for different conversation rounds. On the left, only conversations that have reached the given number of rounds are included. On the right, conversations that end early are padded to 35 rounds through the addition of empty messages/actions.

## C    SCENE SIMILARITY METRIC

Given a ground-truth scene $C$ and a predicted scene $\hat{C}$ (where the presence of a clip art type $c$ in the scene $C$ is indicated by $c \in C$) scene similarity $s$ is defined as:

$$s(C, \hat{C}) = \frac{\sum_{c \in C \cap \hat{C}} f(c)}{|C \cup \hat{C}|} + \frac{\sum_{c_i, c_j \in C \cap \hat{C}, i < j} g(c_i, c_j)}{|C \cup \hat{C}|(|C \cap \hat{C}| - 1)}$$

where

$$
\begin{aligned}
f(c) =\ & w_0 \\
& - w_1 1_{\text{clip art piece } c \text{ faces the wrong direction}} \\
& - w_2 1_{\text{clip art piece } c \text{ is Mike or Jenny and has the wrong facial expression}} \\
& - w_3 1_{\text{clip art piece } c \text{ is Mike or Jenny and has the wrong body pose}} \\
& - w_4 1_{\text{clip art piece } c \text{ has the wrong size}} \\
& - w_5 \sqrt{\left(\frac{\hat{x}_c - x_c}{W}\right)^2 + \left(\frac{\hat{y}_c - y_c}{H}\right)^2}
\end{aligned}
$$

$$
\begin{aligned}
g(c_i, c_j) =\ & - w_6 1_{(\hat{x}_{c_i} - \hat{x}_{c_j})(x_{c_i} - x_{c_j}) < 0} \\
& - w_7 1_{(\hat{y}_{c_i} - \hat{y}_{c_j})(y_{c_i} - y_{c_j}) < 0}
\end{aligned}
$$

Here $x_c$ and $y_c$ refer to the position of the clip art piece in the ground-truth scene, $\hat{x}_c$ and $\hat{y}_c$ refer to its position in the predicted scene, and $W, H$ are the width and height of the canvas, respectively.

We use parameters $w = [5, 1, 0.5, 0.5, 1, 1, 1, 1]$, which provides a balance between the different components and ensures that scene similarities are constrained to be between 0 and 5.

## D    QUALITATIVE EXAMPLES

Figure 9 shows some examples of scenes and dialogs from the CoDraw dataset. The behavior of our Drawer and Teller models on a few randomly-selected scenes is illustrated in Figures 10, 11, and 12.

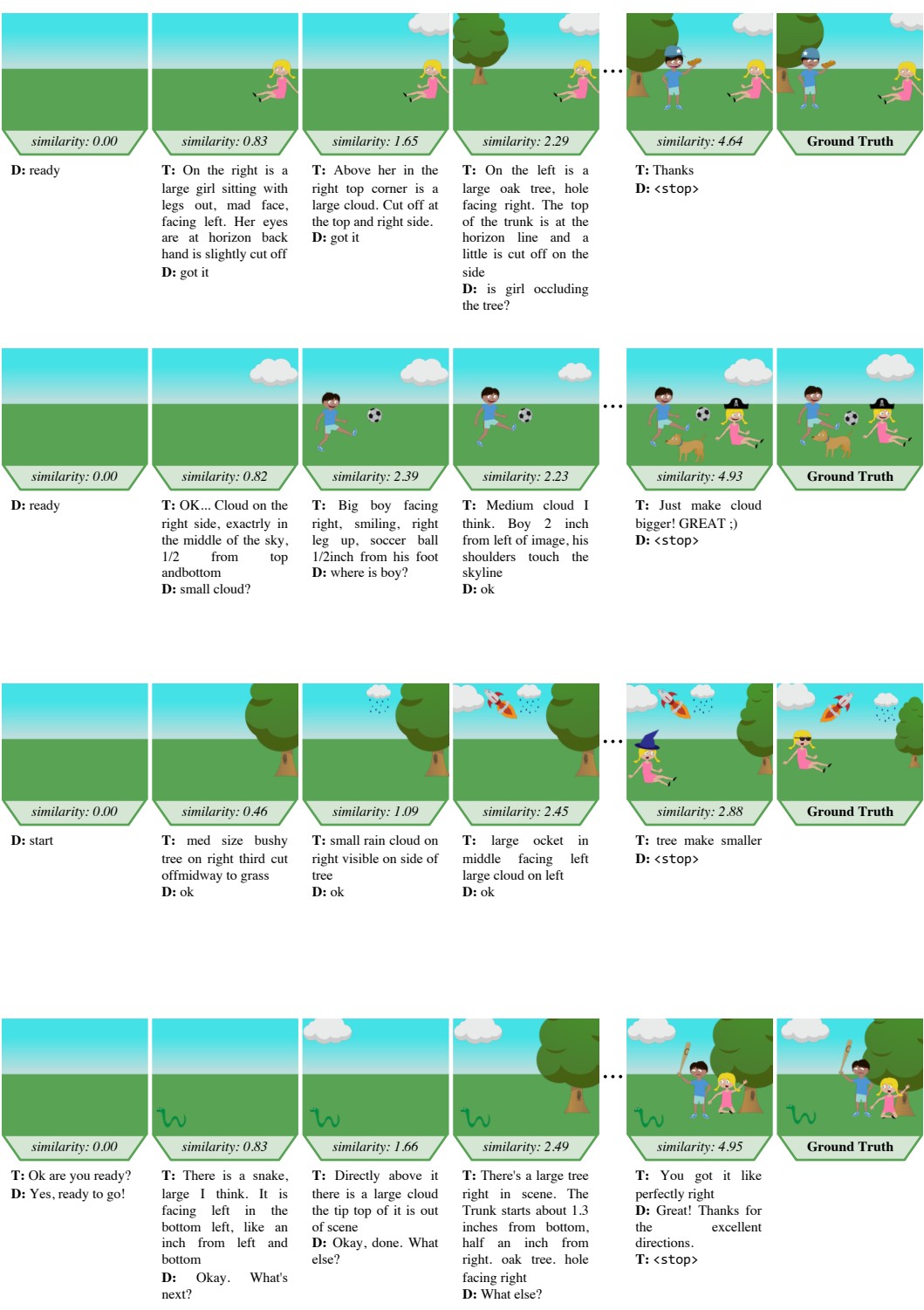

Figure 9: Examples from the Collaborative Drawing (CoDraw) dataset, chosen at random from the test set. The images depict the Drawer's canvas after each round of conversation. From left to right, we show rounds one through four, then the last round, followed by the ground truth scene. The corresponding conversations between the Teller (T) and Drawer (D) are shown below the images. Note that there is no restriction on which of the two participants begins or ends the dialog.

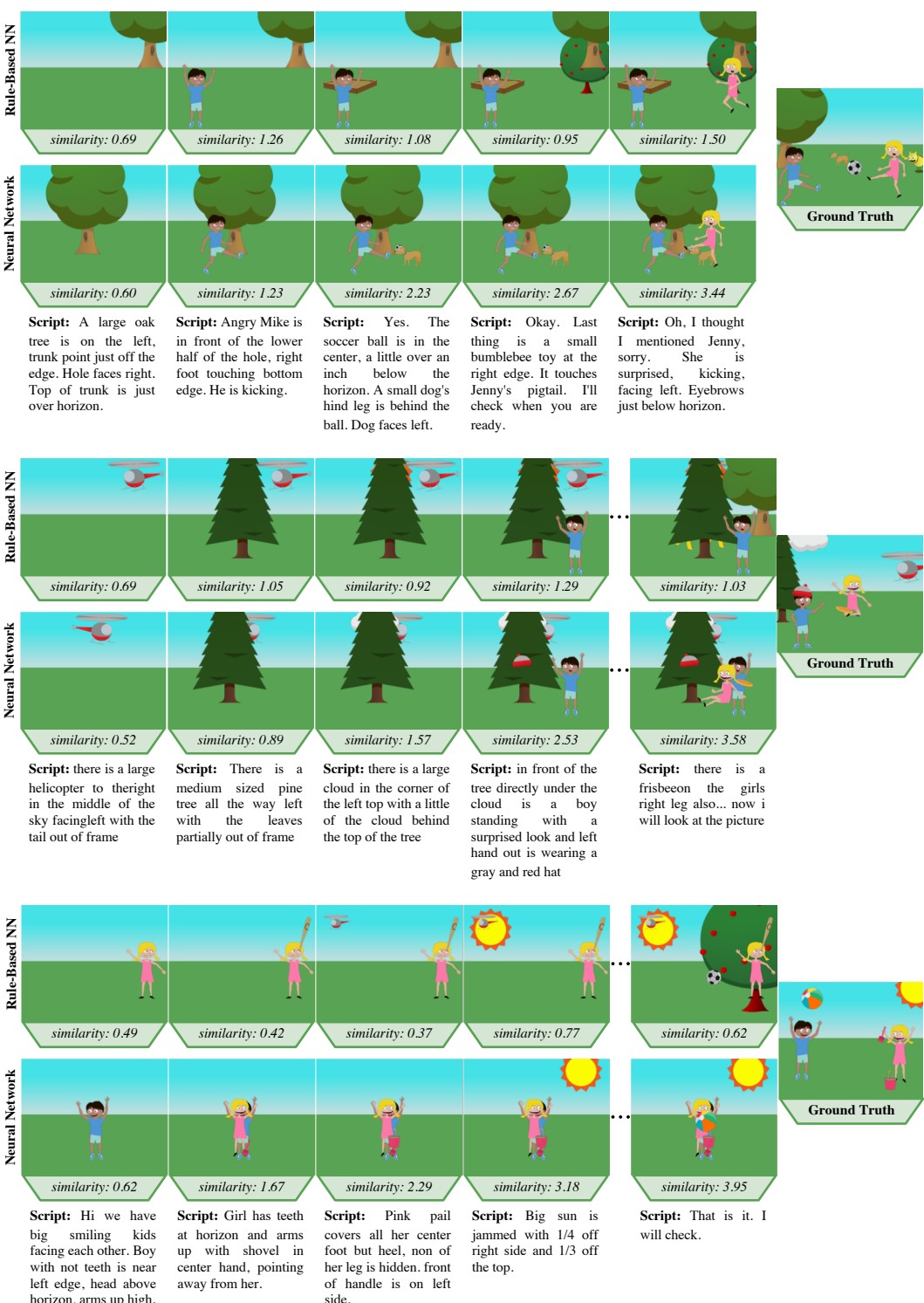

Figure 10: Drawer model behavior where the Teller is simulated by replaying messages associated with three randomly-selected scenes from the test set. The images depict the Drawer's canvas after each round of conversation. For each of the three scenes, the behavior of the Rule-Based Nearest-Neighbor Drawer is shown in the upper row and the behavior of the Neural Network Drawer is shown in the lower row.

Scene A

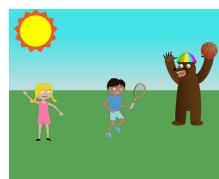

| Scene, Round | Rule-Based Nearest-Neighbor | Scene2seq | | |
| --- | --- | --- | --- | --- |
| | | Imitation learning | + Auxiliary Loss | + RL Fine-tuning |
| A1 | in the top left corner is a medium sun very close to the top and left edge | medium sun on left , 1 2 inch from top and side | on the right hand side is a medium bear facing left with a basketball in his left hand | medium sun top left |
| A2 | guy angry arms down facing right like his right knees touches the top right corner of the table | medium bear on right , facing left , 1 2 " from bottom , small basketball in left hand , racket in left hand | in the middle of the grass is a medium sized boy facing right with a racket in his left hand | medium boy on right angry facing right holding racket |
| A3 | right under it is medium girl , waving with a sad face , facing right . | small bear on right side , facing left , 1 2 " from bottom , small bear on left facing right , armpits on horizon , | he is holding a racket in his left hand and a tennis racket in the middle of the sky | medium girl on left facing right sad one arm up holding basketball |
| A4 | on the right side of screen , medium bear , chest on horizon , paw almost touching right side of screen | small bear on right facing left , chest at horizon , small basketball in left hand , 1 inch from left , 1 2 from bottom | on the left hand side is a medium girl with sad face facing right with a racket in her right hand . | medium bear on right facing left wearing rainbow hat basketball racket in right hand |
| A5 | he also has a rainbow hat , but it 's not snugly on him , it 's only slightly touching his head . | small bear on right facing left , chest at horizon , small basketball in left hand , racket in left hand , racket in left hand , racket in left | the bear is facing left . the bear is facing right . the bear is facing left . | <STOP> |
| A6 | sorry there 's a basketball above the boy 's head near the tree | <STOP> | <STOP> | |
| A7 | she is holding a racket in her left hand . | | | |
| A8 | <STOP> | | | |

Scene B

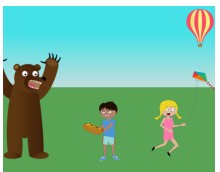

| Scene, Round | Rule-Based Nearest-Neighbor | Scene2seq | | |
| --- | --- | --- | --- | --- |
| | | Imitation learning | + Auxiliary Loss | + RL Fine-tuning |
| B1 | two finger spaces to right of helicopter , small hot air balloon . top of balloon level with helicopter propeller | small hot air balloon , top right , 1 2 " from top , 2 " from right | small hot air balloon on right corner , fully visible . | small hot air balloon top right |
| B2 | in front of tree is boy , he is to the left part of tree and is covering the curve up . he is angry , standing , arms , out facing left | small girl , running , facing right , surprised , 1 " from bottom , 1 2 " from left | small hot balloon on right corner , half " from top . | large bear on left faced right |
| B3 | the head of surprised girl is on front the trunk . she is like running and faces right . | small pine tree behind her , bottom of trunk at horizon , bottom of trunk at horizon , small boy in front of tree , head touching bottom of tree , standing , smiling , facing right , holding a hot dog in left hand | on center , a mad mike with hands front facing left . | medium shocked girl running facing right holding hot dog in right hand |
| B4 | large bear on left side . facing right . arm cut off on left . feet half inch from bottom | he is at the edge , small size , 1 2 " from bottom , 1 2 " from bottom , small kite in left corner , facing right | on center , a small jenny , surprised , running facing right , shoulders at horizon . | boy on right angry facing left holding kite in right hand hot dog in right hand |
| B5 | place a hot dog in her left hand | <STOP> | on center , a small hot balloon , half finger from right side . | <STOP> |
| B6 | right top a kite , small size , tail is off screen . below kite dog medium size , facing left | | a small kite on right corner , only half visible . | |
| B7 | <STOP> | | <STOP> | |

Figure 11: A comparison of the descriptions generated by each of our Teller models for two randomly-sampled scenes from the test set.

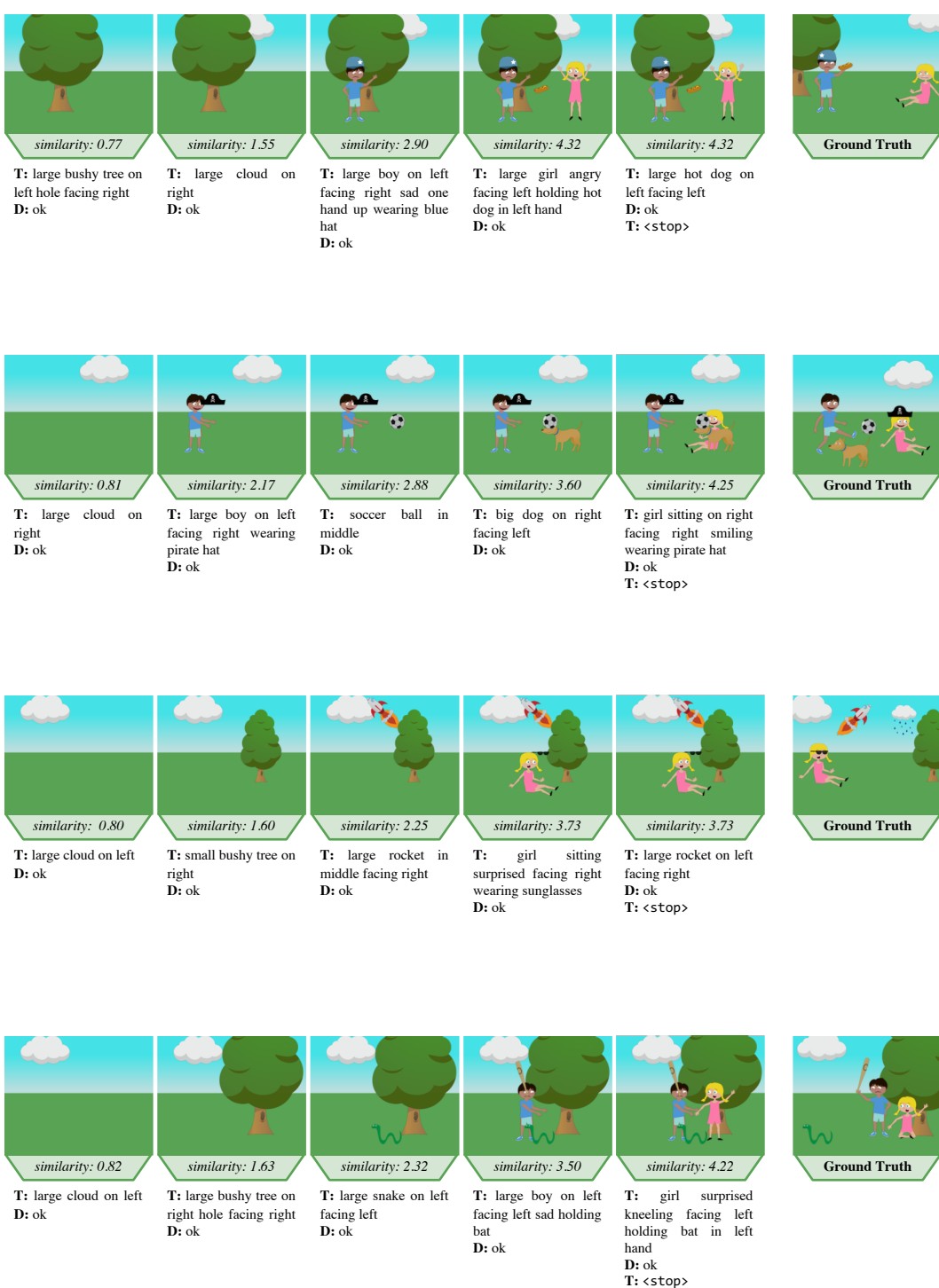

Figure 12: Dialogs from our best Teller model (scene2seq with an auxiliary loss and RL fine-tuning) communicating with our best Drawer model (Neural Network Drawer). The dialogs feature the same scenes as in Figure 9, which were sampled at random from the test set. From left to right, we show the first to the fifth rounds of conversations, followed by the ground truth scene. Our Teller model chose to use exactly five rounds for each of these four scenes. The corresponding conversations between Teller (T) and Drawer (D) are shown below the images.

