# OpenReview forum: "CoDraw: Collaborative Drawing as a Testbed for Grounded Goal-driven Communication"
_ICLR.cc/2019/Conference_

### Official Review · AnonReviewer2 · 2018-11-02
**Exciting task! Not sure about model results**

**Rating:** 7
**Confidence:** 4

**Review:**

This paper presents CoDraw, a grounded and goal-driven dialogue environment for collaborative drawing. The authors argue convincingly that an interactive and grounded evaluation environment helps us better measure how well NLG/NLU agents actually understand and use their language — rather than evaluating against arbitrary ground-truth examples of what humans say, we can evaluate the objective end-to-end performance of a system in a well-specified nonlinguistic task. They collect a novel dataset in this grounded and goal-driven communication paradigm, define a success metric for the collaborative drawing task, and present models for maximizing that metric.

This is a very interesting task and the dataset/models are a very useful contribution to the community. I have just a few comments below:

1. Results:
1a. I’m not sure how impressed I should be by these results. The human–human similarity score is pretty far above those of the best models, even though MTurkers are not optimized (and likely not as motivated as an NN) to solve this task. You might be able to convince me more if you had a stronger baseline — e.g. a bag-of-words Drawer model which works off of the average of the word embeddings in a scripted Teller input. Have you tried baselines like these?
1b. Please provide variance measures on your results (within model configuration, across scene examples). Are the machine–machine pairs consistently performing well together? Are the humans? Depending on those variance numbers you might also consider doing a statistical test to argue that the auxiliary loss function and and RL fine-tuning offer certain improvement over the Scene2seq base model.

2. Framing: there is a lot of work in collaborative / multi-agent dialogue models which you have missed — see refs below to start. You should link to this literature (mostly in NLP) and contrast your task/model with theirs.

References
Vogel & Jurafsky (2010). Learning to follow navigational directions.
He et al. (2017). Learning Symmetric Collaborative Dialogue Agents with Dynamic Knowledge Graph Embeddings.
Fried et al. (2018). Unified pragmatic models for generating and following instructions.
Fried et al. (2018). Speaker-follower models for vision-and-language navigation.
Lazaridou et al. (2016). The red one!: On learning to refer to things based on their discriminative properties.

---

> ### Author Response · Authors · 2018-11-25
> **Re: Exciting task! Not sure about model results**
>
> Thank you for your feedback!
>
> We've updated the related works section to include some of the references you provided and contrast the CoDraw task with these works.
>
> We tried several drawer variations that we did not include in the submission due to space concerns. Replacing the LSTM in the drawer with a bag-of-words representation results in an average score of 3.04 (compared to 3.34 when using an LSTM).  If we additionally remove the dependence on the current state of the canvas — such that the drawer has no memory of prior events in the conversation -- the score drops further to 2.71. Both language understanding and longer-term reasoning are important for the drawer to achieve the performance we report in the paper.

---

### Official Review · AnonReviewer3 · 2018-11-03
**An artificial task for modeling and evaluation of goal-oriented dialogs**

**Rating:** 6
**Confidence:** 4

**Review:**

The paper proposes a game of collaborative drawing where a teller is
to communicate a picture to a drawer via natural language.  The picture
allows only a small number of components and a fixed and limited set
of detailed variations of such components.

Pros:

The work contributed a dataset where the task has relatively objective
criteria for success.  The dataset itself is a valuable contribution
to the community interested in the subject.   It may be useful for
purposes beyond those it was designed for.

The task is interesting and its visual nature allows for easy inspection
of the reasons for successes or failures.  It provides reasonable grounding
for the dialog.  By restricting the scope of variations through the options
and parameters, some detailed aspects of the conversation could be explored
with carefully controlled experiments.

The authors identified the need for and proposed a "crosstalk" protocol
that they believe can prevent leakage via common training data and
the development of non-language, shared codebooks that defeat the purpose
of focusing on the natural language dialog.

The set up allows for pairing of human and human, machine and machine,
and human and machine for the two roles, which enables comparison to
human performance baselines in several perspectives.

The figures give useful examples that are of great help to the readers.

Cons.:

Despite the restriction of the task context to creating a picture with
severely limited components, the scenario of the dialogs still has many
details to keep track of, and many important facets are missing in the
descriptions, especially on the data.

There is no analysis of the errors.  The presentation of
experimental results stops at the summary metrics, leaving many
doubts on why they are as such.

The work feels somewhat pre-mature in its exploration of the models
and the conclusions to warrant publication.  At times it feels like the
authors do not understand enough why the algorithms behave as they do.
However if this is considered as a dataset description paper and
the right expectation is set in the openings, it may still be acceptable.

The completed work warrants a longer report when more solid conclusions
can be drawn about the model behavior.

The writing is not organized enough and it takes many back-and-forth rounds
of checking during reading to find out about certain details that are given
long after their first references in other contexts.  Some examples are
included in the followings.

Misc.

Section 3.2, datasets of 9993 dialogs:
Are they done by humans?   Later it is understood from further descriptions.
It is useful to be more explicit at the first mention of this data collection effort.
The way they relate to the 10020 scenes is mentioned as "one per scene", with a footnote on some being removed.
Does it mean that no scene is described by two different people?  Does this
limit the usefulness of the data in understanding inter-personal differences?

Later in the descriptions (e.g. 4.1 on baseline methods) the notion of
training set is mentioned, but up to then there is no mentioning of how
training and testing (novel scenes) data are created.
It is also not clear what training data include: scenes only?
Dialogs associated with specific scenes?  Drawer actions?

Section 4.1, what is a drawer action?  How many possibilities are there?
From the description of "rule-based nearest-neighbor drawer" they seem to be
corresponding to "teller utterance".
However it is not clear where they come from.  What is an example of a drawer action?
Are the draw actions represented using the feature vectors discussed in the later sections?

Section 5.1, the need for the crosstalk protocol is an interesting observation,
however based on the description here, a reader may not be able to understand
the problem.  What do you mean by "only limited generalization has taken place"?  Any examples?

Section 5, near the end: the description of the dataset splits is too cryptic.
What are being split?  How is val used in this context?

All in all the data preparation and partitioning descriptions need substantial clarification.

Section 6:  Besides reporting averaged similarity scores, it will be useful to report some error analysis.
What are the very good or very bad cases?  Why did that happen?
Are the bad scenes constructed by humans the same as those bad scenes
constructed by machines?  Do humans and machines tend to make different errors?

---

> ### Author Response · Authors · 2018-11-25
> **Re: An artificial task for modeling and evaluation of goal-oriented dialogs**
>
> Thank you for your comments!
>
> We have updated the draft to make it more clear that the contributions of this paper are the new dataset, an associated evaluation protocol, and models that highlight challenges in the dataset as well as will serve as strong baselines for future work on this dataset.
>
> We've added a new Section 6.1 to the paper discussing errors made by our models. These errors reflect the challenging aspects of the CoDraw task. We've also updated the appendix to include a greater variety of qualitative results. The examples there should also help establish a qualitative feel for how the various models differ.
>
> Our updated draft has a new Figure 3 to give an example of codebook-like language use by agents trained on the same data.
>
> We have also re-written several paragraphs (including those that describe data preparation) for clarity based on your recommendations.

---

### Official Review · AnonReviewer1 · 2018-11-06
**Mostly a dataset paper, writing is not coherent, results are not convincing**

**Rating:** 4
**Confidence:** 4

**Review:**

In this paper a new task namely CoDraw is introduced. In CoDraw, there is a teller who describes a scene and a drawer who tries to select clip art component and place them on a canvas to draw the description. The drawing environment contains simple objects and a fixed background scene all in cartoon style. The describing language thus does not have sophisticated components and phrases. A metric based on the presence of the components in the original image and the generated image is coined to compute similarity which is used in learning and evaluation.  Authors mention that in order to gain better performance they needed to train the teller and drawer separately on disjoint subsets of the training data which they call it a cross talk.

Comments about the task:
The introduced task seems to be very simplistic with very limited number of simple objects. From the explanations and examples the dialogs between the teller and drawer are not natural. As explained the teller will always tell ‘ok’ in some of the scenarios. How is this different with a system that generates clip art images based on a “full description”? Generating clip arts based on descriptions is a task that was introduced in the original clip art paper by Zitnick and Parikh 2013. This paper does not clarify how they are different than monologs of generating scenes based on a description.

Comments about the method:
I couldn’t find anything particularly novel about the method. The network is a combination of a feed forward model and an LSTM and the learning is done with a combination of imitation learning and REINFORCE.


Comments about the experimental results:
It is hard to evaluate whether the obtained results are satisfying or not. The task is somehow simplistic since there a limited number of clip art objects and the scenes are very abstract which does not have complications of natural images and accordingly the dialogs are also very simplistic. All the baselines are based on nearest neighbors.

Comments about presentation:
The writing of this paper needs to be improved. The current draft is not coherent and it is hard to navigate between different components of the method and different design choices. Some of the design choices are not experimentally proved to be effective: they are mentioned to be observed to be good design choices. It would be more effective to show the effect of these design choices by some ablation study.
There are many details about the method which are not fully explained: what are the details of your imitation learning method? Can you formalize your RL fine-tuning part with the use of some formulations? With the current format, the technical part of the paper is not fully understandable.

---

> ### Author Response · Authors · 2018-11-25
> **Challenges posed by the CoDraw task**
>
> Thank you for your feedback!
>
> As you point out, the task, dataset, and evaluation protocol are among the main contributions of this work. We also present several models that highlight challenges in the dataset and can serve as strong baselines for future work on this task. We have updated the draft to make it more clear what the contributions are.
>
> There are substantial differences between CoDraw and previous work involving abstract scenes. Here are several:
>
> 1. The need to faithfully reconstruct the entire image results in longer and more detailed descriptions. At the bottom of this comment, we provide an example of language associated with the same scene in different datasets
> 2. Past work has focused mostly on scene generation from sentences. CoDraw, on the other hand, also requires doing the reverse (generating sentences from scenes). Our task is a natural way to "ground" image caption generation into a objective task with measurable evaluation. This is a contribution as well.
> 3. Our dataset records the Drawer's canvas at each step of the dialog. If all we had was a monologue with a single image at the end, we wouldn't be able to build most of the models discussed in this paper.
>
> The task poses a number of challenges:
>
> 1. The Teller must describe the scene in a sensible order. Describing the clip art pieces in a random order would be incoherent and hard to understand. People do this using a combination of planning and incorporating world knowledge: for example, it makes more sense to say "there is a sandbox / the boy sits in the sandbox" than "the boy is in a sitting position / there is a sandbox below him"
> 2. The Teller must describe all aspects of the scene without omitting anything important. Maintaining such long-term coherency is  actually a significant challenge: simply training our LSTM-based Teller to minimize perplexity on the training data results in a model that frequently describes the same objects multiple times while omitting others entirely. As we show in our paper a rule-based nearest-neighbor baseline outperforms the imitation-learning approach for this reason!
> 3. In the example below, the language includes transitions like "next to the swing" and "inside the sandbox" that maintain the flow of the dialog by referring back to previous parts of the conversation. A Teller bot should learn to generate such transitions.
>
> We respectfully disagree with the implication that the clip art domain results in simplistic language. There are many tasks in NLP that use simple domains but real language, for example the SCONE dataset (Long et al. 2016). We're still dealing with noisy and ambiguous text written by humans. Here are a few sentences from CoDraw to give a flavor of some linguistic challenges:
>
> 1. "far left girl chest at skyline reaching hands to  right, happy , on her right smallest cat facing left". The words left/right can be used in multiple ways: "far left" is a position in the absolute frame of reference, "on her right" is a relative position in the girl's frame of reference, and "facing left" indicates direction.
> 2. "in the center is a pine tree with an owl in it and it is wearing a wizards hat.": will a model know to rule out the interpretation where the tree is wearing the hat?
>
> ==========
> Sentences for the same scene, from multiple datasets:
>
> A. Zitnick and Parikh 2013:
> "Mike is upset because his sand castle got destroyed by Jenny's soccer ball."
>
> B. Zitnick et al. 2013:
> 0: Jenny kicked the soccer ball into the sandbox.
> 1: Mike was playing in the sandbox.
> 2: The playground had lots of toys to play with.
>
> C. CoDraw (this work):
>
> T: to the front left side a surprised girl is in the kicking motion. there is a small swing behind her.
> D: ok
> T: next to the swing is a spring bee with a sand box in front of it
> D: ok
> T: inside the sand box is a sad boy and a soccer ball on the corner
> D: ok
> T: that is everything

---

### Meta-Review · Area_Chair1 · 2018-12-20

**Confidence:** 4
**Recommendation:** Reject

**Metareview:**

The reviewers raise a number of concerns including no methodological novelty, limited experimental evaluation, and relatively uninteresting application with very limited real-world application. This set of facts has been assessed differently by the three reviewers, and the scores range from probable rejection to probable acceptance. I believe that the work as is would not result in a wide interest by the ICLR attendees, mainly because of no methodological novelty and relatively simplistic application. The authors’ rebuttal failed to address these issues and I cannot recommend this work for presentation at ICLR.